# Far-Red Component Enhances Paramylon Production in Photoautotrophic *Euglena gracilis*

**DOI:** 10.3390/bioengineering12070763

**Published:** 2025-07-15

**Authors:** Zhaida I. Aguilar-Gonzalez, Anaiza Rico-Luna, Tóshiko Takahashi-Íñiguez, Héctor V. Miranda-Astudillo

**Affiliations:** Departamento de Biología Molecular y Biotecnología, Instituto de Investigaciones Biomédicas, Universidad Nacional Autónoma de México, Mexico City 04510, Mexico; zhaida2001@gmail.com (Z.I.A.-G.); anaizarilu@gmail.com (A.R.-L.); toshikoti@iibiomedicas.unam.mx (T.T.-Í.)

**Keywords:** *Euglena gracilis*, paramylon production, photobioreactor, spectrum optimization, light adaptations, far-red light

## Abstract

In recent years, microalgae have gained significant biotechnological importance as a sustainable source of various metabolites of industrial interest. Among these, paramylon, a polysaccharide produced by the microalga *Euglena gracilis*, stands out for its diverse applications in biomedicine and pharmaceuticals. *E. gracilis* is an adaptable secondary eukaryote capable of growing photoautotrophically, heterotrophically and mixotrophically. During photoautotrophic growth, varying light conditions impact biomass and paramylon production. To investigate the effects of varying illumination more thoroughly, we designed and built a modular photobioreactor that allowed us to simultaneously evaluate the photoautotrophic growth of *E. gracilis* under twelve different light conditions: seven single-spectrum lights (ultraviolet, royal blue, blue, green, red, far-red, and infrared) and five composite-spectrum lights (3000 K, 10,000 K, and 30,000 K white lights, amber light, and “Full-spectrum” light). The 24-day growing kinetics were recorded, and the growth parameters were calculated for each light regime. Both growth curves and pigment composition present differences attributable to the light regime used for cell culture. Additionally, photosynthetic and respiratory machinery functionality were proven by oximetry. Finally, our results strongly suggest that the far-red component enhances paramylon production during the stationary phase.

## 1. Introduction

Population growth inherently generates an increasing energy demand, and approximately 90% of this demand is fulfilled by burning fossil fuels [1]. This activity, along with the economic progress of societies based on industrialization, is the leading cause of the excessive increase in greenhouse gas emissions, with atmospheric CO_2_ being the most abundant, strongly contributing to global warming, which unleashes extreme weather events that destabilize the ecological balance and lead to the destruction of biodiversity [2]. One way to reduce large amounts of atmospheric CO_2_ is through its fixation into organic compounds. In this sense, photosynthetic organisms have great importance, since they are capable of synthesizing nutrients and biomass from CO_2_ and water using light energy. Indeed, the conversion of energy through oxygenic photosynthesis is what sustains all living forms on the planet [3]. From the perspective of the current climate situation, photosynthetic capacity plays an essential role, since microalgae have 10–50 times greater carbon-fixing capacity than terrestrial plants [4,5]. Microalgae are therefore considered important producers of biotechnologically relevant compounds of interest due to their rapid growth, ease of cultivation, and high photosynthetic efficiency, which contributes significantly to biomass production [6]. Furthermore, the generated biomass is an excellent source of biofuels and bioactive metabolites, such as pigments, proteins, fatty acids, and carbohydrates, the synthesis of which is closely related to the growth conditions [7].

*E. gracilis* is one of the most well-characterized organisms in the euglenoid group due to its metabolic versatility and biotechnological potential. It is a free-living, biflagellate, photosynthetic eukaryote with one active emergent flagellum, the other being short and sheltered in a deep pocket or reservoir [8,9]. This species can be grown in heterotrophic, mixotrophic, or photoautotrophic conditions; its ability to adapt to various environments allows for its ubiquitous distribution in freshwater. These adaptations are due to its genetic diversity, which is attributed to its secondary endosymbiotic origin [10]. *E. gracilis* can produce a variety of metabolites, including polyunsaturated fatty acids, provitamin A, vitamins C and E, wax esters, and paramylon, which is a carbohydrate β-1,3-glucan [11,12] that is similar to curdlan synthesized by *Rhizobium* species [13]. Paramylon serves as an energy reserve carbohydrate stored in membrane-bound granules within the cytoplasm. These granules are 1–6 µm long, are surrounded by a biomembrane, and exist in various numbers and species-specific forms [9,11]. The amount of paramylon accumulated in the cells of *E. gracilis* can represent up to 60–70% of its dry weight [14], reaching maximum levels in heterotrophic and mixotrophic cultures via a supplementation of the growth media with carbon sources such as acetate, ethanol, pyruvate, propionate, lactate, succinate, glutamate, and glucose [15,16,17,18,19].

The biomedical applications of paramylon have attracted interest due to its potential as a non-metabolizing fiber that helps reduce obesity, facilitates the healing of acute liver injuries, and exhibits immunostimulatory and antimicrobial activities [17,20,21]. In addition, paramylon hydrogels have been demonstrated to possess anti-inflammatory properties and the ability to accelerate wound repair by promoting angiogenesis [22,23]. One of the main factors affecting the growth and production of compounds in microalgae is the quantity and quality of light [24]. Previous analyses have shown that *E. gracilis* biomass and paramylon production are enhanced during photoautotrophic growth under mixed red and blue light ratios [25]. Considering that only composite spectra have been evaluated, various light conditions still require exploration. The aim of this work was to design and construct a system to assess the biomass and paramylon production of *E. gracilis* by evaluating twelve different light spectra simultaneously: seven single-spectrum lights and five composite-spectrum lights. Furthermore, oxygen evolving capacity, cellular respiration, and paramylon production were determined under each light condition.

## 2. Culture System Design

Figure 1A presents a schematic illustration of the *Ankaa* Photobioreactor System (PBR). This PBR consists of twelve compartments measuring 11.5 cm × 15.5 cm × 14.5 cm (length × width × height), designed to culture twelve 25–125 mL Erlenmeyer flasks simultaneously. Each compartment is equipped with (1) a chiller plate to maintain a constant temperature inside each flask, (2) an air-based heat sink and (3) a specific light-emitting diode (LED) module. A 24 L water tank connected to a C-250 water chiller (Boyu) regulates the temperature (16–35 °C) of the entire system. Chilled water recirculates (1400 L/h) through each heat sink plate located under the culture flask with the assistance of a SP-2500 water pump (Boyu). The heat dissipation system features 4 cm × 4 cm fans located in the upper polycarbonate cover at the center of each compartment. Additionally, a fan is positioned next to each LED to extract any heat generated by the LED module (Figure 1A). Finally, the light system features twelve different 3 W LED modules which can be classified as either single- or composite-spectrum (Figure 1B). Seven compartments contain single-spectrum lights distributed throughout the visible light spectrum: ultraviolet (UV, emission peak: 395 nm), royal blue (RB, emission peak: 437 nm), blue (B, emission peak: 460 nm), green (G, emission peak: 520 nm), red (R, emission peak: 635 nm), far red (FR, emission peak: 733 nm), and infrared (IR, emission peak: 850 nm) (Figure 1B, upper panel). The other five compartments contain the composite-spectrum lights: 3000 K (3K), 10,000 K (10K) and 30,000 K (30K) white lights, amber light (A), and “Full-spectrum” light (F) (Figure 1B, lower panel).

## 3. Material and Methods

### 3.1. Culture Conditions and Growth Curves

*Euglena gracilis* (SAG 1224-5/25) was obtained from the University of Göttingen (Sammlung von Algenkulturen, Göttingen, Germany). Cells were grown under continuous light conditions, illuminated with the corresponding LED module at an intensity of 10 μmol photons m^−2^ s^−1^. Cells were cultured using 125 mL Erlenmeyer flasks containing 50 mL of liquid Tris-minimum-phosphate medium (TMP) at pH 7.0, supplemented with a vitamin mix (biotin 10^−7^%, B12 vitamin 10^−7^%, and B1 vitamin 2 × 10^−5^% *w*/*v*) at 20 °C. Flasks were manually homogenized three times a day. The culturedevelopment was monitored in a Neubauer double-ruled counting chamber. A volume of 10 μL of the cell suspension was loaded into the chamber, and counting was performed under a microscope (Olympus CH2, Hachioji, Japan) with an EA 40× objective. Growth data were adjusted to a Gompertz growth model using GraphPrism 8.0.2 software, and maximal growth rate and doubling time were calculated accordingly.

### 3.2. Pigment Quantification

Two-milliliter samples (~8 × 10^5^ cells) were collected from each culture during the logarithmic growth phase, and room-temperature absorbance spectra were recorded using a Cary 6000 UV-vis spectrophotometer (Agilent, Santa Clara, CA, USA). After harvesting the cells, 1 mL of absolute methanol was added, and the pellet was vortexed. The pellet was then incubated in an ice bath in complete darkness for 12 h. After this period, centrifugation was performed at 6700× *g*/10 min to separate the insoluble material, the supernatant was recovered, and room-temperature absorbance spectra were measured. The amounts of chlorophyll *a*/chlorophyll *b* and total carotenoids were determined in accordance wit the protocol followed by Ritchie [26] and Caspers [27], respectively.

### 3.3. Oxygen Evolution and Oxygen Consumption

Oxygen production was measured under each light condition using the corresponding growth light with a Clark-type oxygen electrode (Oxygraph+, Hansatech Instruments Ltd., King’s Lynn, UK), as previously described [28]. In short, 1.5 × 10^7^ cells were harvested in their logarithmic phase by centrifugation (2200× *g*/5 min) and resuspended in 1 mL of fresh culture medium (TMP) supplemented with 2 mM NaHCO_3_ from a freshly prepared 20 mM stock solution and filtered by a 0.22 µm PVDF membrane. All measurements were carried out at 25 ± 1 °C and followed for 10 min. Rates for mitochondrial respiration were measured over a 10 min period by dark incubation after oxygen production measurements. Finally, the cell suspension was recovered and Chl *a* was quantified as described above.

### 3.4. Paramylon Quantification

Extraction and total paramylon quantification were determined as described by Rodríguez-Zavala et al. [19] with slight modifications. Briefly, 7.6 × 10^5^ cells from the stationary culture phase were collected by centrifugation at 2200× *g*/20 min; 1 mL of 1% SDS was added to the pellet, which was vortexed for 1 min prior to a 15 min incubation in boiling water. Then, the samples were incubated in an ice bath for 10 min, the paramylon pellet was recovered by centrifugation at 2200× *g*/20 min, and the supernatant was discarded. This step was repeated twice with an additional 1% SDS wash. The clean paramylon pellets were resuspended in 1 mL of NaOH 1 M and vortexed. Samples measuring 40 μL were transferred to a clean tube, followed by the addition of 600 μL of phenol 5% and 2.5 mL of 95% H_2_SO_4_. The samples were incubated for 20 min at 25 °C, and the absorbance at 490 nm was determined. The concentration of paramylon was determined against a calibration curve (0–100 μg) of dextrose (USP standard).

### 3.5. Brightfield Microscopy

Life cell imaging of R and FR cells in the stationary phase was recorded with a Microscope Digital Camera (MU1603, AmScope, Irvine, CA, USA) using a microscope (Olympus CH2) with an EA 40× objective. For chloroplast highlighting, the image was processed digitally in AmScope 4.11 software by modifying hue parameter to −88 in HSL color mode for the whole image.

### 3.6. Statistical Analysis

A total of three biological replicates were analyzed. One-way ANOVA tests were performed followed by a post hoc Dunnett’s multiple comparisons test between each condition against the rest. For statistical significance, *p*-value thresholds were defined as ns: 0.1234, *: 0.0332, **: 0.0021, ***: 0.0002, and ****: <0.0001. Statistical analysis was performed using GraphPrism software. Raw data from the full statistical analysis are presented in the Appendix A; for clarity, the *p*-value threshold of 0.05 was used to define statistical significance in the graphs presented in the main text.

## 4. Results

### 4.1. Light Wavelength as an Environmental Factor Affecting Euglena gracilis Growth

In order to evaluate the light adaptation capacity of *E. gracilis*, we tested phototrophic growth using twelve different light conditions. An inoculum of 7 × 10^5^ 3K-acclimated cells was transferred to fresh medium and grown under seven single-wavelength and five composite-spectrum lights. Twenty-four-day growth kinetics were followed for each condition. Typical three-phased growth curves were observed for most of the conditions (Figure 2A,B); the logarithmic growth phase encompassed days 5 to 14 for the UV, RB, B, FR, 3K, 10K, 30K, A, and F light regimes, while it was shifted between days 7–20 and after day 20, for the R and G light conditions, respectively. For the G condition, the 24-day kinetics were not sufficient to reach the stationary phase, and no detectable growth could be achieved with the IR light regime. Among the single-spectrum lights, RB and FR were associated with the largest biomass production (above 8.5 × 10^5^ cells/mL) followed by B light (~7.5 × 10^5^ cells/mL), while the UV and R regimes led to the 3.5 × 10^5^ cells/mL value being exceeded at the start of the stationary phase (Figure 2A). The largest cell density was reached with the 3K, 10K and F composite spectra (above 9 × 10^5^ cells/mL), while the A and 30K light regimes led to the 7.0 × 10^5^ cells/mL value being exceeded at the start of the stationary phase (Figure 2B). Growth parameters, i.e., maximal growth rate and doubling time, were calculated. The larger maximal growth rate values were observed under UV, RB and FR light (Figure 2C) and the 10K, 30K and F composite-spectrum conditions (Figure 2E), followed by the B, R and A spectrum conditions (Figure 2C). Thus, the doubling time under red light was the largest (3.64 ± 0.27 days) among all the tested light regimes (Figure 2D,F).

### 4.2. E. gracilis Adapts Its Pigment Composition as an Acclimatization Strategy to Specific Light Regimes

To obtain insights into the light adaptation consequences in *E. gracilis*, we characterized the pigment composition of cells grown under each light regime. Room-temperature absorption spectra indicate slightly different pigment contents between each growth condition (Figure 3A,E). Interestingly, the spectra from cells adapted to the single-wavelength B, G, and R conditions present differences in the Soret band (~480 nm) compared to those under UV, RB and FR conditions (Figure 3A, dark arrowhead). Likewise, the absorption valley between 530 and 560 nm was deeper in the R- and B-adapted cells (Figure 3A, purple arrowhead). Additionally, a red-shifted absorption signal above 710 nm was observed mainly in the FR condition, in contrast to the UV and R spectrum conditions (Figure 3A, red arrowhead). In contrast, no difference in absorption above 710 nm was distinguishable when comparing composite-spectrum-adapted cells (Figure 3E, red arrowhead). In this way, the difference in the Soret band is only noteworthy for 3K-adapted cells (Figure 3E, dark arrowhead); the 530–560 nm absorption valley was deeper under the A and 3K regimes (Figure 3E, purple arrowhead). To further explore these differences in pigment composition, chlorophyll *a* (Chl *a*), chlorophyll *b* (Chl *b*) and total carotenoid content were determined (Figure 3B,F). No notable difference was observed between the single-wavelength (Figure 3B) and composite-spectrum treatments (Figure 3F). In the same way, the Chl *a*/*b* ratios were similar among all light treatments (Figure 3C,G). Total carotenoid quantification revealed that the content of these pigments was distributed between 16 and 22% among all the samples without any striking difference (Figure 3D,H), with the UV condition presenting the largest carotenoid accumulation.

### 4.3. E. gracilis Photosystem II Can Use Different Wavelengths for Oxygen Evolution

To evaluate photosynthetic capacity, especially regarding the photosystem II functionality of the specific-light-acclimated cells, oxygen evolution was measured in the middle of the logarithmic phase using the same light conditions under which each culture had grown (Figure 4A). With the exception of IR, all the tested light spectra promoted oxygen production when irradiated with their corresponding LED. Moreover, mitochondrial functionality was tested as O_2_ consumption by the electron transport chain (mitochondrial respiration) when cells were incubated in the dark (Figure 4B). The composite spectrum 30K along with the single spectrum G presented the highest rates of oxygen production, followed by the composite spectra F, A, and 10K, and the single spectra B and FR. Finally, the least effective lights for O_2_ production were R, 3K, and RB (Figure 4C,E). Error bars in Figure 4C–F represent the average rate of O_2_ production or consumption within the first three minutes, during which the photosynthetic oxygen evolution and respiration were monitored. In this context, it is evident that while some light spectra, such as 3K, UV, and RB, show only minor deviations, other light regimes exhibited large fluctuations in the rate of O_2_ production over time. Also, mitochondrial respiration rates were the highest for the composite spectra 10K, 30K and A along with the single spectra UV, G and R, followed by the 3K and F composite spectra and the RB, B and FR single spectra (Figure 4D,F). Overall, both photosynthesis and respiration machinery were functional and there was a net production of O_2_ in all light regimes.

### 4.4. Far-Red Light Improves Paramylon Production in Photoautotrophically Grown E. gracilis

Paramylon production (pg/cell) was determined from cells grown in TMP minimal medium under twelve light regimes during the stationary phase (Figure 5). Paramylon production in cells grown under single-spectrum light resulted in values between 72 and 134 pg/cell, with the highest value recorded under FR light (Figure 5A). Regarding the quantity of paramylon obtained in cells grown under composite-spectrum light during the stationary phase, the values ranged from 79 to 122 pg/cell, with the highest values found under the A and F light conditions, the emission spectra of which shifted above 680 nm (Figure 1B). The condition in which the highest paramylon production was obtained was that with the FR spectrum, reinforcing the fact that the >680 nm light component enhances paramylon production (Figure 5C). To qualitatively assess the differences in paramylon production among live cells, representative photomicrographs were taken from *E. gracilis* grown in R (Figure 6A–C) and FR (Figure 6D–F). Brightfield microscopy makes evident that using FR to grow cells augments both the number and size of paramylon granules (Figure 6D–F) compared to those when R is used to grow cells (Figure 6A–C).

## 5. Discussion

Short-term light adaptations and long-term light acclimations are complex processes that ensure the survival of photosynthetic organisms under changing environmental conditions. Traditionally, photosynthetically active radiation (PAR) was defined as being between 400 and 700 nm [29], but the development of light-emitting diodes (LEDs) that emit monochromatic light has allowed for the study of photosynthetic activity using narrow spectra [30]. Our in-house designed *Ankaa* PBR (Figure 1) allowed us to evaluate the simultaneous effect of 12 different light regimes on the growth and paramylon production of *Euglena gracilis* in a simple but reliable manner. Recently, it has been shown that *E. gracilis* can grow under different ratios of red:blue wavelengths [25]. This is in line with our observation of the capacity of this microorganism to grow photoautotrophically using light with slightly variant composite spectra (Figure 2B), which are mainly composed of a mixture of blue/red lights (Figure 1B). Indeed, our results showed that composite spectra enriched with blue components (10K, 30K and F) strongly contribute to higher cell densities compared to those where the blue component is lower (3K and A) (Figure 2B). Additionally, our results evidenced the great degree of flexibility of this species with regards to growth when irradiated with a single wavelength (Figure 2A). Similar acclimation capacity to a wide diversity of specific wavelengths has been observed in other microalgae like *Chlamydomonas reinhardtii* [31], *Scenedesmus* sp. [32] and *Nannochloropsis* sp. [24]. Interestingly, our data suggest that this species survives infrared radiation, although no detectable growth was observed under the IR regime throughout the 25-day experiment. Nevertheless, subsequent transfer of the IR-irradiated cells reilluminated under 3K light conditions showed visible culture growth. This adaptation capacity was described in several diatom species a long time ago [33] but, to our knowledge, this is the first observation of *E. gracilis* surviving for a long period without an extended-PAR regime (395–750 nm) in the absence of any carbon source.

Through evolution, genetic drift, and speciation, the light-harvesting machinery of microalgae has given rise to adaptation mechanisms that enable these organisms to thrive in diverse environments. Chlorophyll-based light harvesting complexes (LHCs) absorb blue and red light efficiently, although in the green gap (between 500 and 600 nm), their absorption is rather low; this is in line with the cell absorption profiles found among all the evaluated light regimes (Figure 3A,B, purple arrow heads). Green light has been proposed as a suitable growth condition in highly dense PBR due to its augmented penetrability compared to that of blue or red light [34,35]. Additionally, it has been shown that green wavelengths are required for optimal growth in many microalgal species [36]. Furthermore, in the chlorophyte *Picochlorum* sp. [37], green light (510 nm) is sufficient for the efficient growth of cultures that exhibit a two-day prolonged initial lag phase and chemical modifications in chlorophyll *a* and chlorophyll *b* characterized by an unsaturation in the phytol side chain. Similarly, our results showed that *E. gracilis* can grow under a green light regime, with the presence of a 12-day prolonged initial lag phase compared to the other light regimes (Figure 2A). Nevertheless, further research is needed to characterize any fine-tuned pigment modifications in this growth condition.

Light utilization by photosynthetic organisms depends not only on its pigment content but also on the organization of those pigments in LHCs. In this way, absorption spectrum analysis can provide valuable information about the arrangement of the photosynthetic machinery in living cells. In *E. gracilis*, the principal pigments are chlorophyll *a* and diadinoxanthin, while chlorophyll *b* content is low [38]. Recently, a detailed description of the *E. gracilis* photosynthetic apparatus has been used to explain the far-red light absorption capacity of the species-specific mobile antenna LHCE [39]. Moreover, this study highlights the adaptation mechanisms used by *E. gracilis* to compensate for low light intensity. A larger chlorophyll *a*/*b* ratio (Figure 3C,G) may indicate an increase in the relative abundance of LHCE complexes as a photoacclimation strategy which aligns with the presence of a red-shifted absorption signal (Figure 3A,E red arrow heads), as could be the case for G and R regimes (Figure 3C). However, this LHCE expression was not enough to compensate for the unfavorable light conditions of the G and R regimes, as observed through the diminished biomass accumulation and growth rates (Figure 2A,C). On the other hand, when far-red light is used, the increase in the LHCE complex allowed the photosynthetic machinery to harvest more light (Figure 3A), improving cellular growth (Figure 2A) as well as paramylon production (Figure 5). Moreover, carotenoid biosynthesis and accumulation in *E. gracilis* are regulated by different intensities of blue or red light [40]. These photosynthetic pigments protect against photoexcitation from the excess light absorbed by the LHCs [41] and stabilize the structural integrity of these LHCs [42]. Our pigment determination for the *Euglena* cells from all the evaluated light regimes showed only a slight increase in total carotenoids in UV-treated cells in comparison to the rest of the conditions (Figure 3D), and this carotenogenic effect was also observed in *Coccomyxa* sp. and *Chromochloris* sp. irradiated with UV-A [43].

For optimal photosynthesis, it is imperative to maintain balanced excitation in both photosystems [29], which depends on the rate of photon absorption and the efficiency of conversion of light into chemical energy. Even though a highly efficient excitation energy is achievable at a particular wavelength, the photosynthetic efficiency can still be limited by the rate of carbon fixation [44]. The oxygen-evolving complex within the PSII catalyzes the reduction of water to oxygen. Since the main pigments in PSII, Chl *a* and Chl *b*, better capture light in the red and blue wavelengths, higher oxygen production rates are expected at those wavelengths. Thus, the 30K, A, and F light regimes, which have a blue and red component, had the highest oxygen production rates, along with G lights (Figure 4A,C). The crosstalk between chloroplastic photosynthesis and mitochondrial respiration in photosynthetic eukaryotic cells is crucial for optimum cell metabolism. During the different light treatments, the oxygen production rates were in concordance with the observed mitochondrial respiration rates (Figure 4B,D). An interesting exception was seen in the R light- and FR light-adapted cells: even though the PSII oxygen evolution rate was low under R light, the respiration rate was higher, and vice versa for the FR light treatment. A similar increase in photosynthetic CO_2_ assimilation due to the far-red component was described previously for soybean [45]. The evaluation of oxygen evolution (Figure 4A,C) and mitochondrial respiration (Figure 4B,D) indicated that both bioenergetic organelles maintained fully functional energy pathways under all our treatments.

Biotechnologically, *E. gracilis* is particularly interesting for the direct conversion of CO_2_ into biomass under several stress conditions, including acidic media, high salt concentrations and the presence of heavy metals [46,47]. Paramylon biosynthesis in photoautotrophic conditions only uses the excess energy captured by the cells [48] and requires aerobic conditions [49]. Because our cultivation conditions involved continuous-light regimes, paramylon production was not limited. Nevertheless, biomass accumulation (Figure 2A) and paramylon production were lower under the UV, G, and R light regimes (Figure 5A), indicating restrictions to the available energy under these conditions. Contrastingly, the red light regime (620–680 nm) induced starch accumulation in *Chlamydomonas* sp. [50] and *Chlorella vulgaris* without affecting photosynthetic efficiency [51]. On the other hand, when light utilization was optimal (RB, B, and FR light and composite-spectrum light), paramylon production increased notably (Figure 5A,B). According to the emission spectra determined for each LED (Figure 1B), only the amber (A), “full spectrum” (F), and far-red (FR) regimes had a component above 680 nm (Figure 5C), which is in concordance with paramylon production enhancement. Therefore, we propose that this light component above 680 nm contributes to paramylon production during the stationary phase. In agreement with previous reports, cells grown under FR light exhibited the largest paramylon production [16], which gave rise to larger paramylon granules directly visible under brightfield microscopy [52] (also see Figure 6).

## 6. Conclusions

Taken together, our data reinforce the idea that *E. gracilis* possesses light adaptation capabilities exhibiting robust plasticity. This characteristic allows it to grow and produce important metabolites under several divergent light regimes. Interestingly, paramylon production was enhanced by the far-red light component during the stationary phase. Our findings pave the way to further explore fine structural modifications in *E. gracilis* photosynthetic machinery under different illumination conditions. In addition, the flexible light utilization exhibited by this species shows that it occupies a privileged position as a producer of metabolites of biotechnological interest and that it may also contribute to alleviating the current climate crisis.

## Figures and Tables

**Figure 1 bioengineering-12-00763-f001:**
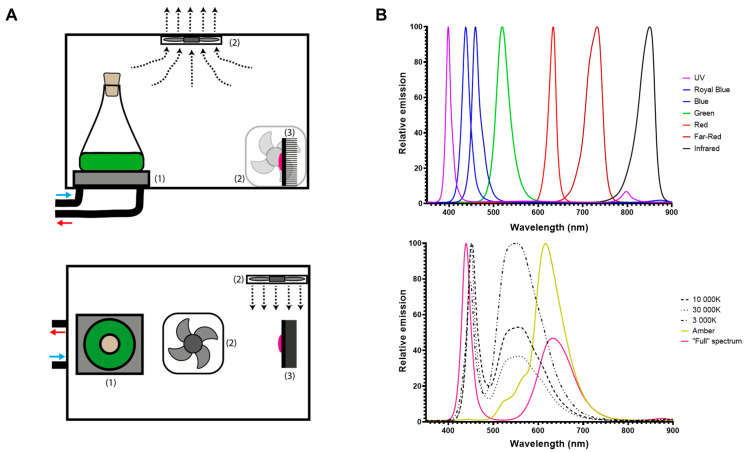
Schematic representation of *Ankaa* photobioreactor system. (**A**) Diagram of one compartment seen laterally (upper panel) or from above (lower panel). Each compartment is composed of the following: (1) a chiller plate, (2) a heat dissipation system, (3) an LED module, and a 25–125 mL Erlenmeyer flask. Arrows indicate the entrance (blue) and exit (red) of chilled water. (**B**) Relative emission spectra from twelve different 3 W LED modules from *Ankaa*, separated into single spectra (upper panel) and composite spectra (lower panel).

**Figure 2 bioengineering-12-00763-f002:**
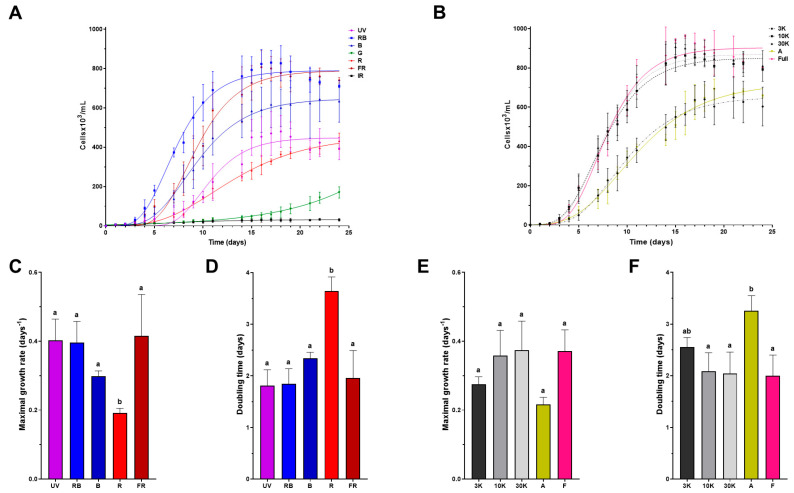
Effect of different light spectra on *Euglena gracilis* growth. Growth curves under single- (**A**) and composite-spectrum light (**B**); growth data were adjusted to a Gompertz growth model. Maximal growth rate (**C**,**E**) and doubling time (**D**,**F**) are depicted, with error bars representing SEM for single- (**C**,**D**) and composite-spectrum light (**E**,**F**). Values with the same letter in the column indicate that there was no significant difference between those conditions (*p* < 0.05).

**Figure 3 bioengineering-12-00763-f003:**
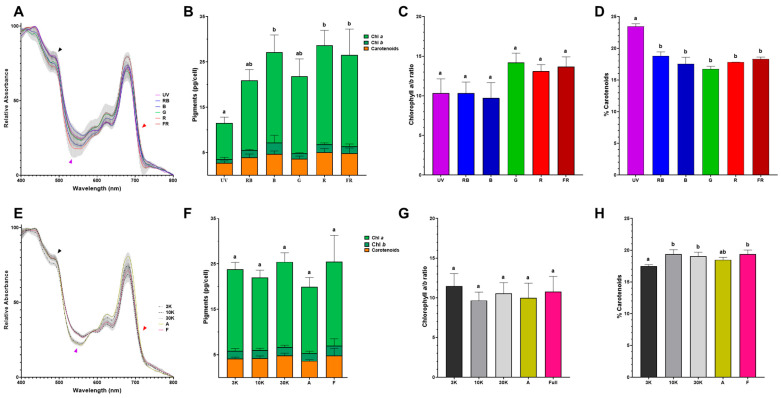
Differences in pigment composition due to growth light conditions. Normalized room-temperature absorption spectra of *Euglena gracilis* growth under single- (**A**) and composite-spectrum lights (**E**). The arrowheads indicate differences in absorption spectra between conditions (see Section 4.2 for details). Total pigment distribution for single- (**B**) and composite-spectrum lights (**F**). Chlorophyll *a*/*b* ratio (**C**,**G**) and carotenoid percentage (**D**,**H**) for single- (**C**,**D**) and composite-spectrum light (**G**,**H**). Error bars represent SEM for all panels. Values with the same letter in the column indicate that was no significant difference between the conditions (*p* < 0.002).

**Figure 4 bioengineering-12-00763-f004:**
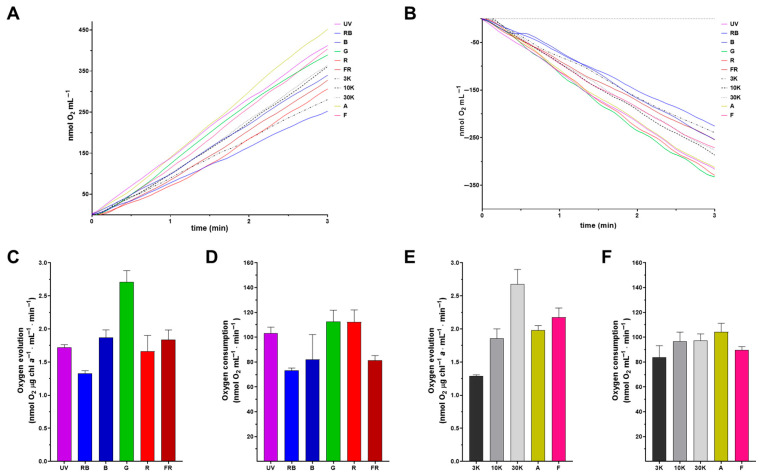
Oxygen evolution and dark respiration in *Euglena gracilis* acclimated to distinct light regimes. Oxygen evolution (**A**) and dark respiration (**B**) of phototrophic cells (15 × 10^6^) grown under distinct light spectra. Average O_2_ production (**C**,**E**) and consumption per minute (3 min) (**D**,**F**). Error bars represent the SDs within the first three minutes.

**Figure 5 bioengineering-12-00763-f005:**
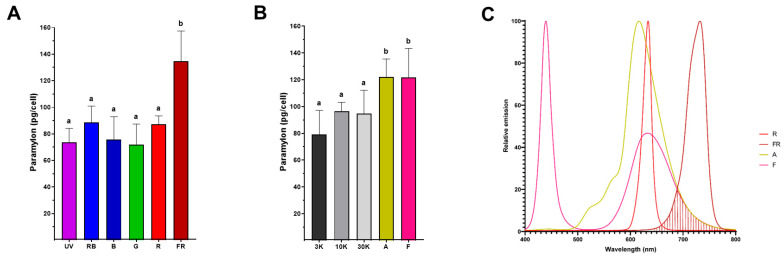
Paramylon production in the stationary growth phase under single- (**A**) and composite-spectrum light (**B**). Error bars represent SEM. Values with the same letter in the column indicate that there was no significant difference between the conditions (*p* < 0.002). The far-red component (>680 nm) is present among the A, F and FR light regimes (**C**).

**Figure 6 bioengineering-12-00763-f006:**
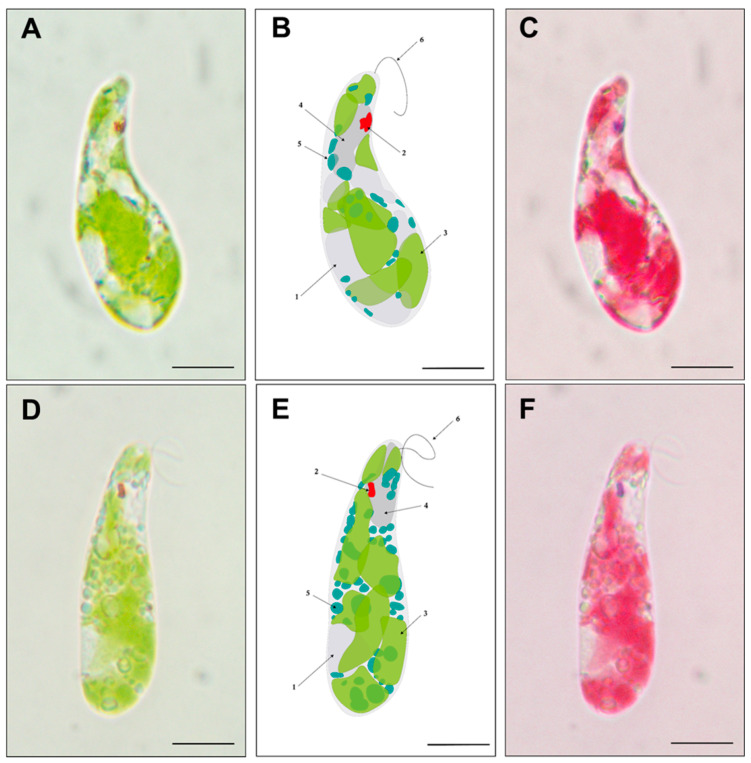
Paramylon accumulation in *Euglena gracilis*. Representative microphotography of cells containing paramylon granules grown under R (**A**–**C**) and FR (**D**–**F**) light regimes. Seemingly, both the quantity and size of paramylon granules are augmented when cells are grown under FR light. Bright-field microscopy of *E. gracilis* (**A**,**D**) and its schematic representation (**B**,**E**), where the following distinct cell elements are indicated by arrows: (1) the nucleus, (2) eyespot, (3) chloroplast, (4) vacuole, (5) paramylon granule, and (6) flagellum. The digital process of bright-field microscopy for chloroplast highlighting (**C**,**F**). The scale bar represents 10 μm.

## Data Availability

Data is contained within the article or Appendix A.

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
