# Peer review of "Far-Red Component Enhances Paramylon Production in Photoautotrophic Euglena gracilis"

_bioengineering, 2025, doi:10.3390/bioengineering12070763_

Round 1
Reviewer 1 Report
Comments and Suggestions for Authors
In this manuscript, the authors designed a photobioreactor that allowed the simultaneous assessment of photoautotrophic growth of E. gracilis under twelve different light conditions. In my opinion, the manuscript is good, and the authors presented a good amount of data. I think it is suitable for publication after some revisions.
Comments
The presentation of the results of the study is missing in the abstract.
How is the culture mixed? The light distribution on the cells depends directly on how the culture is mixed.
References should be checked for correctness since the references 29 and 44 cite the same article “Zhen, S.; van Iersel, M. W.; Bugbee, B. Photosynthesis in sun and shade: the surprising importance of far-red photons. New Phytologist 2022, 236: 538–546. https://doi.org/10.1111/nph.18375”.
Minor comments:
Replace “x” in equations with the symbol “×”.
Author Response
Comment #1
The presentation of the results of the study is missing in the abstract.
Answer to comment #1
We thank the reviewer´s advice, accordingly, we modified the abstract section to emphasize the results (lines 21-25).
Comment #2
How is the culture mixed? The light distribution on the cells depends directly on how the culture is mixed.
Answer to comment #2
We added in the Material and Methods section lines 115-116 the following clarification: “Flasks were manually homogenized three times a day.”
Comment #3
References should be checked for correctness since the references 29 and 44 cite the same article “Zhen, S.; van Iersel, M.W.; Bugbee, B. Photosynthesis in sun and shade: the surprisingimportance of far-red photons. New Phytologist 2022, 236: 538–546. https://doi.org/10.1111/nph.18375”.
Answer to comment #3
We thank the reviewer´s advice, accordingly, we checked and corrected the references in text and in the list of references.
Comment #4
Replace “x” in equations with the symbol “×”.
Answer to comment #4
We replaced “x” in equations with the symbol “×” in all the manuscript.
Reviewer 2 Report
Comments and Suggestions for Authors
This manuscript presents a well-structured and comprehensive investigation into the effects of varying light spectra on the growth and metabolite production of Euglena gracilis. The experimental design is robust, and the methods are clearly and accurately described. The study includes an extensive dataset collected over a 24-day cultivation period, including growth kinetics, pigment composition, photosynthetic performance, and paramylon content. The authors have applied appropriate statistical methods to support their data analysis, which strengthens the validity of their conclusions. The findings offer valuable insights into optimizing microalgal cultivation strategies for biotechnological applications. The work is relevant and clearly presented.
I recommend acceptance of the manuscript in its current form.
Author Response
Comment #1
This manuscript presents a well-structured and comprehensive investigation into the effects of varying light spectra on the growth and metabolite production of Euglena gracilis. The experimental design is robust, and the methods are clearly and accurately described. The study includes an extensive dataset collected over a 24-day cultivation period, including growth kinetics, pigment composition, photosynthetic performance, and paramylon content. The authors have applied appropriate statistical methods to support their data analysis, which strengthens the validity of their conclusions. The findings offer valuable insights into optimizing microalgae cultivation strategies for biotechnological applications. The work is relevant and clearly presented.
I recommend acceptance of the manuscript in its current form.
Answer to comment #1
We appreciate the comments about our work.
Reviewer 3 Report
Comments and Suggestions for Authors
Summary:
In this manuscript, the authors investigated the influence of diverse light conditions on Euglena gracilis growth and paramylon production by designing and utilizing a custom-designed photobioreactor. They characterized and analyzed growth kinetics, pigment composition, photosynthesis, respiration, and paramylon production for different light regimes. Based on their results and interpretations, different light conditions exerted various influences on physiological activities. E. gracilis showed a capacity to adapt to varying spectral environments. Paramylon accumulation was phase-specifically varied. Exposure to far-red light appeared to enhance the paramylon synthesis during the stationary phase.
General concept comments:
The topic is of biotechnological significance given the increasing industrial interest in microalgal-derived products such as paramylon. The experimental approach of designing a modular photobioreactor and applying twelve distinct light regimes represents a practical approach for exploring light-dependent physiological responses. The aims are well-aligned with the data presented, and the overall design and presentation of this study are relatively standard. The manuscript would benefit from more thorough in-depth analysis and interpretation and clearer descriptions of methodologies, which would enhance both its scientific contribution and its relevance to potential industrial applications.
Specific comments:
- In the Materials and Methods section, line 117 states that the cells were grown at 20 °C, whereas line 140 mentions that the measurements of oxygen evolution and oxygen consumption were performed at 25 ± 1 °C. Could the authors please clarify why different temperatures were used, and why the measurements were not consistently conducted at the growth temperature of 20 °C?
- In Figure 2A, under green (G) and red (R) light conditions, it appears that Euglena gracilis had not yet entered the stationary phase. Was growth monitored over a longer time span to fully capture the growth curve under these light conditions? If not, further clarification is needed on how the growth rate and doubling time were calculated under such conditions. Moreover, in Figure 5A/C, paramylon production during logarithmic and stationary phases was compared under different light conditions. However, based on Figure 2, the timing of reaching the stationary phase appears to vary substantially between conditions. Thus, it would be helpful and necessary to extend the duration of the growth profile.
- For Figure 3A/E, could the authors specify the parameters used for normalization of the room temperature spectra?
- The manuscript would benefit from a clearer explanation of the significance of absorption spectra measurements in characterizing growth and other physiological activities of the organism.
- Figure 5A shows that under RB and B light conditions, paramylon production during the logarithmic phase was higher than under FR light. However, in lines 276–277, it is stated that “Among both growth phases, the condition in which the highest paramylon production was obtained was FR spectrum,” which appears to lack precision and may require clarification.
- Regarding the composite light spectra, the manuscript should provide more specific explanations and discussion on how they reinforce the conclusion that light components above 680 nm enhance paramylon production. Additionally, it is noteworthy that paramylon production does not significantly increase under RB and B light during the stationary phase (compared to other light conditions), while under R and G light conditions it reaches a level comparable to that under B light. Further interpretation, discussion, or relevant literature references (if necessary and applicable) are recommended for this observation.
- For Figure 6 and the description in lines 278–282 regarding the use of microscopic imaging to “evaluate the differences in paramylon production among live cells”, the methodology section should include more details on how microscopy was employed for this evaluation, including image acquisition, processing, and quantification methods for determining “both the number and size of paramylon granules” (Lines 281-282).
- The observations and “conclusions” presented in lines 316–319 are very interesting. However, it is unclear why the related data and results were not included or further discussed. Providing these results would strengthen the “conclusions”.
- Finally, the manuscript would be significantly improved by further discussion on the physiological relevance and potential interrelationship among the main characterizations: growth, photosynthesis, respiration, and paramylon production.

Author Response
Comment #1
In the Materials and Methods section, line 117 states that the cells were grown at 20 °C, whereas line 140 mentions that the measurements of oxygen evolution and oxygen consumption were performed at 25 ± 1 °C. Could the authors please clarify why different temperatures were used, and why the measurements were not consistently conducted at the growth temperature of 20 °C?
Answer to comment #1
We agree with the reviewer 's comment regarding there is a difference in temperature between cell growth and oxygen-related measurements. Our scope was to compare the oxygen evolution and oxygen consumption at the same temperature among all the light regimes. In our lab, we routinely measure mitochondrial respiration at controlled-room temperature (25 ± 1), and according to Dongsansuk et al., 2013 (DOI: 10.1007/s11099-012-0070-2), the quantum yield of PSII photochemistry, remained remarkably unchanged over a broad temperature range (5-45 °C) in plants, therefore, we consider that the expected oxygen evolution rate won´t vary significantly between 20 and 25°C, in this way we can compare both parameters (production and consumption) at same temperature, and these determinations can be further compared against parallels projects.
Comment #2
In Figure 2A, under green (G) and red (R) light conditions, it appears that Euglena gracilis had not yet entered the stationary phase. Was growth monitored over a longer time span to fully capture the growth curve under these light conditions? If not, further clarification is needed on how the growth rate and doubling time were calculated under such conditions. Moreover, in Figure 5A/C, paramylon production during logarithmic and stationary phases was compared under different light conditions. However, based on Figure 2, the timing of reaching the stationary phase appears to vary substantially between conditions. Thus, it would be helpful and necessary to extend the duration of the growth profile.
Answer to comment #2
For all the light conditions tested the maximal growth rate and doubling time parameters were calculated with the data obtained from the 24-day growth kinetics. According to the Gompertz growth model adjustment (Figure 2A/B), only the G condition did not allow us to calculate maximal growth rate and doubling time parameters. To clarify this point, we added the next sentence in lines 185-186 “For G condition the 24-day kinetics was not sufficient to reach the stationary phase.” As reviewer suggested, longer time is necessary to complete the growth curve in G regime, nevertheless, this is beyond the scope of our manuscript, since we aimed to characterize the process of early light acclimation.
Growth parameters were recalculated directly from the adjustment to the Gompertz model equation as described in Tjørve and Tjørve 2017 (DOI: 10.1371/journal.pone.0178691) and Figure 2 was updated.
With respect to paramylon quantification, we agree that the comparison of carbohydrate production between logarithmic and stationary phases raises deeper questions, therefore, to avoid misleading in the main conclusion of our manuscript, we removed the determination of paramylon production in the logarithmic phase, and focus our findings in the production on stationary phase. We updated Figure 5 and main text, accordingly.
Comment #3
For Figure 3A/E, could the authors specify the parameters used for normalization of the room temperature spectra?
Answer to comment #3
The normalization used in all the spectra presented among the manuscript, i.e. emission or absorption (Figures 1B, 3A/E and 5C), corresponds to adjust maximal and minimal values as 100 and 0, respectively, according to the following formula: Normalized value = (Value - Minimal)/(Maximal - minimal).
We changed the scale of the y-axis from 0-1 to 0-100 in Figure 3A/E, to maintain the same normalization units as Figures 1B and 5C.
Comment #4
The manuscript would benefit from a clearer explanation of the significance of absorption spectra measurements in characterizing growth and other physiological activities of the organism.
Answer to comment #4
Considering that composition of PSI and PSII core complexes has been well conserved during the evolution (Caffarri et al., 2014, DOI: 10.2174/1389203715666140327102218), the pigment content, together with absorption spectra, broadly describe the antenna quantity in photosynthetic organisms. Recently, two types of antenna complexes have been described in Euglena gracilis: conserved LHCII antenna complex and specie-specific LHCE antenna complex, the main difference between them is the chlorophyll content and their spectroscopic properties, LHCE complex is mainly composed of chlorophyll a, while LCHII is enriched with chlorophyll b. Additionally, the LHCE complex presents a remarkable red-shifted light absorption capacity (Miranda-Astudillo et al., 2025, DOI: 10.1101/2025.05.07.652572). Therefore, a larger chlorophyll a/b ratio (Figures 3C/G) indicates the increase of the relative abundance of LHCE complexes as a photoacclimation strategy, this is in line with the presence of red-shifted absorption signals (Figures 3A/E red arrow heads).
Taken together, these peculiar adaptation mechanisms in E. gracilis try to compensate for unfavorable light conditions, i.e. low light intensity or specific non-useful light spectrum (Green or Red spectrum), rather than an adaptation to growth under high light intensities as photoprotection like other species. Therefore, in the case of green and red light regimes, E. gracilis probably compensates these unfavorable conditions expressing more LHCE complexes, as judged by the larger chl a/b ratio (Figure 3C) but this LHCE expression is not enough to compensate the unfavorable conditions, as observed in the growth curves (Figure 2A). On the other hand, when far-red light is used, the increase in the LHCE complex allows the photosynthetic machinery to harvest more light (Figure 3A) and improve cellular growth (Figure 2A).
We added some lines in the discussion section (lines 335-351) to reinforce this point.
Comment #5
Figure 5A shows that under RB and B light conditions, paramylon production during the logarithmic phase was higher than under FR light. However, in lines 276–277, it is stated that “Among both growth phases, the condition in which the highest paramylon production was obtained was FR spectrum,” which appears to lack precision and may require clarification.
Answer to comment #5
We thank reviewer 3 for this observation, according to “answer to reviewer 3 comment #2”, to avoid misleading the conclusion of our manuscript, we removed the determination of paramylon production in the logarithmic phase.
Comment #6
Regarding the composite light spectra, the manuscript should provide more specific explanations and discussion on how they reinforce the conclusion that light components above 680 nm enhance paramylon production.
Answer to comment #6
According to emission spectra determined for each led (Figure 1B), among composite spectra lights, only amber (A) and “full spectrum” (F) present considerable emission above 680 nm, in the same line, during stationary phase paramylon production in these light regimes is comparable with the one observed in Far-red (FR) regime, which is also the only single spectra that presents emission above 680 nm. Therefore, we propose that this light component above 680 nm, which is shared between A, F and FR leds, contributes to enhance paramylon production at stationary phase.
We added some lines in the discussion section (lines 388-392) to reinforce this point.
Comment #7
Additionally, it is noteworthy that paramylon production does not significantly increase under RB and B light during the stationary phase (compared to other light conditions), while under R and G light conditions it reaches a level comparable to that under B light. Further interpretation, discussion, or relevant literature references (if necessary and applicable) are recommended for this observation.
Answer to comment #7
We thank the reviewer for this particular point signaling. Yes, we agree about the difference in the rise of paramylon production between logarithmic phase and stationary phase among light regimes, especially, between RB and B light compared to UV, R and G lights. As mentioned in “answer to reviewer 3 comment #2”, we removed the determinations of paramylon production during the logarithmic phase to avoid misleading the main conclusion of the manuscript, and focus the observations in paramylon production during the stationary phase.
Comment #8
For Figure 6 and the description in lines 278–282 regarding the use of microscopic imaging to “evaluate the differences in paramylon production among live cells”, the methodology section should include more details on how microscopy was employed for this evaluation, including image acquisition, processing, and quantification methods for determining “both the number and size of paramylon granules” (Lines 281-282).
Answer to comment #8
With the brightfield microscopy we intended to have a visual and qualitative approach to the organization of paramylon granules in live cells since gross quantification of paramylon content was obtained as stated in point 3.4 and represented in Figure 5. Further research is needed to quantitatively assess the size and number of paramylon granules such as Transmission Electron Microscopy of isolated granules.
To clarify this point, the following modifications were made to the manuscript.
1) In Methodology section we added (lines 159-164):
3.5 Brightfield microscopy
Life cell imaging of R and FR cells on stationary phase was recorded with a Microscope Digital Camera (MU1603, AmScope) using a microscope (Olympus CH2) with an EA 40x objective. For chloroplast highlighting the digital image was processed digitally in AmScope software by modifying the Hue parameter to -88 in HSL color mode in the whole image.
2) In Figure 6 description (line 287) we changed the statement “Both the quantity and size of paramylon…” for “Seemingly, both the quantity and size of paramylon…”
3) In the results sections (line 273) we changed the statement “To microscopically evaluate the differences in paramylon production among live cells.” for “To qualitatively assess the differences in paramylon production among live cells”
Comment #9
The observations and “conclusions” presented in lines 316–319 are very interesting. However, it is unclear why the related data and results were not included or further discussed. Providing These results would strengthen the “conclusions”.
Answer to comment #9
We agree with the reviewer´s observation regarding that conclusions presented in lines 316-319 open more questions beyond the scope of the present work. We rephrased the corresponding lines in the text to avoid misleading.
Comment #10
Finally, the manuscript would be significantly improved by further discussion on the physiological relevance and potential interrelationship among the main characterizations: growth, photosynthesis, respiration, and paramylon production.
Answer to comment #10
We appreciate the reviewer´s comments, we agree with them. We reanalyzed our growth data, to improve the calculation of growth parameters, and redraw Figures 2C-F and text accordingly. About growth phases we redraw Figure 5 to remove the quantification of paramylon production in logarithmic phase to avoid misleading the main conclusion, and together with the rest of reviewer´s comments we changed several lines in the text.
Round 2
Reviewer 3 Report
Comments and Suggestions for Authors
Dear Authors,
Thank you for the revisions. Most of the previous concerns have been addressed, and the overall clarity and scientific significance of the manuscript have been improved.